# Kinetic Analysis of the Thermal Decomposition of Iron(III) Phosphates: Fe(NH_3_)_2_PO_4_ and Fe(ND_3_)_2_PO_4_

**DOI:** 10.3390/ijms21030781

**Published:** 2020-01-25

**Authors:** Isabel Iglesias, José A. Huidobro, Belén F. Alfonso, Camino Trobajo, Aránzazu Espina, Rafael Mendoza, José R. García

**Affiliations:** 1Departamentos de Física, Universidad de Oviedo, 33007 Oviedo, Spain; iis@uniovi.es (I.I.); mbafernandez@uniovi.es (B.F.A.); 2Departamentos de Matemáticas, Universidad de Oviedo, 33007 Oviedo, Spain; 3Departamentos de Química Orgánica e Inorgánica, Universidad de Oviedo-CINN, 33006 Oviedo, Spain; ctf@uniovi.es (C.T.); jrgm@uniovi.es (J.R.G.); 4Servicios Científico-Técnicos, Universidad de Oviedo, 33006 Oviedo, Spain; maespina@uniovi.es (A.E.); mendozarafael@uniovi.es (R.M.)

**Keywords:** metal phosphates, hydrothermal synthesis, crystal structure, thermal decomposition, kinetics

## Abstract

The hydrothermal synthesis and both the chemical and structural characterization of a diamin iron phosphate are reported. A new synthetic route, by using *n*-butylammonium dihydrogen phosphate as a precursor, leads to the largest crystals described thus far for this compound. Its crystal structure is determined from single-crystal X-ray diffraction data. It crystallizes in the orthorhombic system (*Pnma* space group, *a* = 10.1116(2) Å, *b* = 6.3652(1) Å, *c* = 7.5691(1) Å, *Z* = 4) at room temperature and, below 220 K, changes towards the monoclinic system *P*2_1_/*n*, space group. The in situ powder X-ray thermo-diffraction monitoring for the compound, between room temperature and 1100 K, is also included. Thermal analysis shows that the solid is stable up to ca. 440 K. The kinetic analysis of thermal decomposition (hydrogenated and deuterated forms) is performed by using the isoconversional methods of Vyazovkin and a modified version of Friedman. Similar values for the kinetic parameters are achieved by both methods and they are checked by comparing experimental and calculated conversion curves.

## 1. Introduction

In recent years, the structures and properties of open-framework transition metal phosphates have attracted the interest of many researchers because of their possible utilization in different areas such as catalysis, adsorption, ion exchange and battery electrodes [1,2]. The olivine–phosphate LiFePO_4_ paved the way for a new class of materials as an alternative to cathodes for rechargeable lithium batteries [3]. To date, the interest in this kind of compounds continues due to its potential technical applications, such as electric vehicles and grid storage [4,5]. Moreover, the possibility of different oxidation states and coordination geometries of iron means that iron phosphates exhibit interesting magnetic properties [6,7]. These properties, together with their biocompatibility, make them potentially usable as nanostructured carriers in nuclear medicine. This provides an alternative to lanthanide phosphates, hydroxyapatite or iron oxides (magnetite and maghemite), which have been extensively studied in the field of biomedical applications as nanoconstructs and can be used as core platforms for attaching medical radionuclides [8]. Hydrothermal synthesis conditions play an important role in the products obtained because of the influence of the precursors, molar ratio between reactants, pH, reaction time, etc. [9]. For instance, the direct synthesis of iron(III) phosphates under mild hydrothermal conditions by the reaction of FeCl_3_(aq) and H_3_PO_4_(aq) in the presence of urea [10] showed that the compounds obtained depend on the presence and the concentration of urea in the reaction system. Without urea or at low urea concentrations, only FePO_4_·2H_2_O is formed and as the concentration of urea and the pH in the reaction media increase, the sequential formation of NH_4_Fe(HPO_4_), NH_4_Fe_2_(PO_4_)_2_(OH)·2H_2_O, and Fe(NH_3_)_2_PO_4_ take place.

The polycrystalline diamin iron(III) phosphate, Fe(NH_3_)_2_PO_4_, and its deuterated phase Fe(ND_3_)_2_PO_4_ had been previously synthesized and structurally characterized in our laboratory [11,12]. The nuclear diagram obtained from the powder neutron diffraction data (PND) of the deuterated sample between 30 and 300 K, complementary to the powder X-ray diffraction (PXRD), allowed us to analyze in depth the structural phase transition: orthorhombic (*Pnma* space group) crystal structure at room temperature and monoclinic structure (*P*2_1_/*n* space group) below 226(5) K [12]. PND experiments between 1.8 and 30 K revealed a magnetic phase transition at the temperature of 22 K towards a helimagnetic arrangement with an incommensurate propagation vector ${\vec{k}}_{inc}=(1/2-\tau ,0,\tau )$, being τ≈0.04 rlus (reciprocal lattice units). The magnitude of the Fe(III) magnetic moment is close to $\mu_{Fe}=4.5$ μ_B_ at *T* = 2 K [13].

This paper reports the hydrothermal synthesis of Fe(NH_3_)_2_PO_4_ with an adequate size of crystallites to address the first structural characterization of this material from single-crystal X-ray diffraction. At room temperature, the compound shows orthorhombic symmetry, *Pnma* space group. It is well known that the size and shape of crystals and aggregates are closely related to their properties and may affect their applications. As a function of crystal size, the thermal decomposition of the hydrogenated and deuterated samples has been analysed and its kinetic parameters computed by using isoconversional methods. The values of the activation energy have been obtained as a function of the extent of conversion and the results were checked by getting a model of the kinetic process to obtain the theoretical *α − T* curves and comparing them with the experimental data.

## 2. Results

### 2.1. Morphological and Structural Results

The influence of the hydrothermal synthesis conditions on the final product is well known. In our case, one of the precursors, the phosphoric acid H_3_PO_4_ [10], was replaced by the amine phosphate C_4_H_9_NH_3_(H_2_PO_4_), which affected the crystallization and morphology of the final product. Using this new hydrothermal route, the diamin iron(III) phosphate large crystals were carried out. The scanning electron microscopy (SEM) image of Fe(NH_3_)_2_PO_4_ displayed in Figure 1 suggests the morphology of the sample is constituted by fibres forming compact blocks, unlike polycrystalline phases (hydrogenated and deuterated) [11,13].

The crystal structure of Fe(NH_3_)_2_PO_4_ was determined from single-crystal X-ray diffraction data at room temperature, showing orthorhombic symmetry, *Pnma* (No. 62) space group. The unit cell parameters are *a* = 10.1116(2) Å, *b* = 6.3652(1) Å, *c* = 7.5691(1) Å, *α* = *β* = *γ* = 90° and *Z* = 4. A summary of crystallographic data and refinement parameters is listed in Table 1.

Final atomic coordinates and isotropic displacement parameters are reported in Table 2 and the selected bond distances and angles of the monocrystalline compound are listed in Table 3.

At room temperature, the structure of the compound can be described by isolated iron octahedrons which are sharing corners with four phosphate tetrahedrons and two ammonia molecules in relative cis position (Figure 2).

The ammonia molecules are placed in the channels along the *b* axis. There is only one crystallographic independent iron per unit cell. In Figure 3, a projection of the monocrystalline structure down the *b* axis is shown.

The evolution of the cell parameters with temperature was analysed from the sequential refinement of the profiles obtained from the single-crystal X-ray diffraction to selected temperatures, between room temperature and 2 K, by using the Rietveld method and the FullProf program. The orthorhombic crystal changes towards monoclinic *P*2_1_/*n* structure when the temperature is lowered down to 240 K. The cell parameters, *a* = 6.355 Å, *b* = 10.097 Å and *c* = 7.581 Å, remain constant up to 2 K. The angle *β* of the unit cell varies from 90°, at 250 K, to 92.5°, at 100 K (see Figure 4). Although symmetry is lost, the basic structural characteristics are kept because of strong hydrogen bonds.

The in situ HT-PXRD (300–1100 K) patterns suggest, in Figure 5, that the sample is stable up to 440 K. The ammonium evacuation finishes at ca. 680 K. After this temperature, an intermediate phase occurs until ca. 940 K, when the α-FePO_4_ is formed [14].

### 2.2. Thermogravimetric and Kinetic Analysis

The TG-DTG profiles in an inert atmosphere for Fe(NH_3_)_2_PO_4_ (large crystals), depicted in Figure 6, show that the material is stable up to ca. 440 K. Then it decomposes in three steps with an ammonium evacuation up to ca. 680 K, giving a total weight loss of 18.5% (calc. 18.4%) in the formation of FePO_4_. According to the weight loss of 9.6% (calc. 9.2%) and DTG peaks position, the first step, up to ca. 560 K, can be attributed to the evacuation of one ammonium molecule per formulae. The weight loss of 5.9% (calc. 6.1%) accompanying the next step, up to ca. 610 K, amounts to the evacuation of 2/3 ammonium molecules. In the latest step, up to ca. 680 K, the weight loss, 3.0% (calc. 3.1%), closely corresponds to 1/3 of ammonium molecules.

The determination of the kinetic parameters of the thermal decomposition of Fe(NH_3_)_2_PO_4_ (large crystals) was performed from thermogravimetric data obtained in the temperature range from 298 to 1173 K using several heating rates: 1.25, 2.5, 5, 10 and 20 K min^−1^. The Modified Friedman (MFR) and Vyazovkin (Vyaz) isoconversional methods were applied to evaluate the activation energy as a function of the extent of conversion with an increment ∆α = 0.01. The kinetic analysis was also carried out for Fe(NH_3_)_2_PO_4_ [11] and Fe(ND_3_)_2_PO_4_, hydrogenated and deuterated forms with a smaller crystal size, which had been previously synthesised [12]. The dependence of the activation energy on the extent of conversion is similar for all samples: there are three different zones that occur in the same conversion range (Figure 7).

For the large crystals, in the 0.05 < α < 0.50 conversion range, the apparent activation energy *E* is about 100 kJ mol^−1^, concerning the first step of ammonium loss. For 0.50 < α < 0.70, the values obtained for *E* are about 110 kJ mol^−1^, which correspond to the second step of ammonium loss and finally, in the 0.70 < α < 0.90 conversion range, *E* is about 100 kJ mol^−1^, related to the third step of ammonium loss. The shape of the *E − α* curves is slightly less smooth in the second and third *α*-range for the polycrystalline samples (Figure 7b,c). The difference may be due to a partial overlap of the last two ammonium losses. The values of the activation energy computed by the MFR and Vyaz methods hardly show variations.

In order to model the kinetic process, the product *Af*(*α*) was computed for the mentioned heating rates, by using the obtained values of *E*. The kinetic parameters were checked by reconstructing the *α − T* curves and comparing with the experimental data. As can be observed in Figure 8, a good agreement for both MFR and Vyaz methods has been achieved.

A numerical measure of the difference between the reconstructed and experimental curves is given by $R^{2}=1-SSE/SST$ where $SSE=\sum_{i=1}^{n}\left(\alpha_{i}^{*}-\alpha_{i}\right)/n$ and $SSE=\sum_{i=1}^{n}{\left(\alpha_{i}-\bar{\alpha }\right)}^{2}/n$ being $\alpha_{i}$ the experimental data, $\alpha_{i}^{*}$ the computed values and $\bar{\alpha }=\sum_{i=1}^{n}\alpha_{i}/n$ [15]. The obtained values are: 0.999557, 0.992676, 0.999179, 0.997144 and 0.998747 for MFR method, and 0.998384, 0.992278, 0.999071, 0.996493 and 0.998995 for Vayz method, for *β* = 1.25, 2.5, 5, 10 and 20 K min^−1^, respectively.

## 3. Materials and Methods

### 3.1. Sample Preparation and Analytical Procedures

The synthesis of Fe(NH_3_)_2_PO_4_ large crystals was carried out by a hydrothermal route in a stainless steel, Teflon-lined vessel under autogenous pressure from a mixture of C_4_H_9_NH_3_(H_2_PO_4_) [16], FeCl_3_·6H_2_O (1 M, Merck), and (NH_2_)_2_CO (solid, Merck) in the molar ratio 1:1:16. The total volume of the reaction mixture was 15 mL, and the autoclave was sealed and heated at 453 K for six days. The solid product was filtered off, thoroughly washed with an excess of ionized water until reaching a neutral pH, and dried in air at room temperature. The phosphorus and iron contents of the solid were determined by inductively coupled plasma mass spectrometry (ICP-MS) analysis (Finnigan, Element model) after dissolving a weighed amount in HF(aq). Microanalytical data for nitrogen were obtained with a Perkin Elmer 2400B elemental analyser.

### 3.2. X-ray Diffraction Studies

Powder X-ray diffraction (PXRD) patterns were recorded on a Panalytical X’pert PRO MPD X-ray diffractometer with PIXcel detector, operating in the Bragg–Brentano (θ/2θ) geometry, using CuKα radiation (λ = 1.5418 Å). Data were collected at room temperature between 5 and 80 ° in 2θ with a step size of 0.02° and counting time of 10 s per step. In the X-ray thermodiffraction (HT-PXRD) studies, each powder pattern was recorded in the 10–60° range (1 h) with a scan step size of 0.0131° and a counting time of 0.424 s per step. Temperature intervals of 20 K, from room temperature up to 1073 K, were chosen. The temperature ramp between two consecutive temperatures was 10 K min^−^^1^.

Data collection of single crystal X-ray diffraction (SXRD) was performed at 298 K on an Oxford Diffraction Xcalibur Nova single-crystal diffractometer, using CuKα radiation. Images were collected at a 65 mm fixed crystal-detector distance, using the oscillation method, with 1° oscillation and variable exposure time per image (10–60 s). Data collection was analysed with the program CrysAlis Pro CCD [17]. Data reduction and cell refinement were performed with the program CrysAlis Pro RED (Oxford Diffraction Ltd., 2008). The unit cell dimensions were determined from 4187 reflections between θ = 4° and 74°, and multiple observations were averaged, Rmerge = 0.02, resulting in 538 unique reflections, of which 529 were observed with I > 2σ(I). An empirical absorption correction was applied using the SCALE3 ABSPACK algorithm as implemented in the program CrysAlis Pro RED [17]. The crystal structure was solved by direct methods, using the program SIR-92 [18], and anisotropic least-squares refinement was carried out with SHELXL-97 [19]. All non-hydrogen atoms were anisotropically refined while hydrogen atoms were located in a Fourier difference map and then isotropically refined, riding on their parent atom.

### 3.3. Thermal Analysis and Kinetic Data

A Mettler–Toledo TGA/SDTA851e was used for the thermal analysis in a dynamic nitrogen atmosphere (50 mL min^−^^1^) at several heating rates: 1.25, 2.5, 5, 10, 20 K min^−^^1^. In all cases, ca. 20 mg of powder sample was thermally treated, and blank runs were performed. In thermogravimetric analysis, for each time t, the mass was measured and the extent of conversion $\alpha =(m_{0}-m_{t})//\left(m_{0}-m_{\infty }\right)$ was computed, where $m_{t}$ denotes the mass at time t, and $m_{0}$ and $m_{\infty }$ the sample mass at the beginning and at the end, respectively.

The kinetics of heterogeneous condensed phase reactions, in conditions far from equilibrium and assuming the temperature dependence is given by the Arrhenius equation [20,21], can be described by the general equation
(1)$$\frac{d\alpha }{dt}=A\exp\left(-\frac{E}{RT}\right)f(\alpha )$$
where *t* (s) is the time, *A* (s^−1^) is the pre-exponential factor, *T* (K) is the temperature, *E* (J mol^−1^) is the activation energy, *R* (J mol^−1^ K^−1^) is the gas constant and *f*(α) is the reaction model function. For a non-isothermal process with a constant heating rate *β*, *T* = *T*_0_ + *βt*, Equation (1) can be written as
(2)$$\beta \frac{d\alpha }{dT}=A\;f(\alpha )\exp\left(-\frac{E}{RT}\right)$$

Isoconversional methods are based on the isoconversional principle which states that the reaction rate, at a given extent of conversion, is only a function of temperature [22]. A large number of computational methods have been developed to perform the kinetic analysis that can be achieved without determining the pre-exponential factor or the model function [21]. One of the most popular and simplest methods is that proposed by Friedman [23], which is included in the family of isoconversional differential methods. For a given value of the extent of conversion, and for several runs with different constant heating rates, *β*_i_, *i* = 1,⋯, *n*, taking logarithms in Equation (2), one obtains
(3)$$\ln\left[\beta_{i}{\left(\frac{d\alpha }{dT}\right)}_{\alpha ,i}\right]=\ln\left[A_{\alpha }f(\alpha )\right]-\frac{E_{\alpha }}{RT_{\alpha ,i}}$$

Then, the value of activation energy can be determined from the slope of the plot of $\ln\left[\beta_{i}{\left(d\alpha /dT\right)}_{\alpha ,i}\right]$ against $1/T_{\alpha ,i}$. A general drawback of the differential methods is that they are very sensitive to experimental noise and tend to be numerically unstable, especially when the rate is estimated by numerical differentiation, as happens when thermogravimetric analysis is performed [21]. A modified version of the Friedman method was proposed by Huidobro et al., which is less sensitive to noise effects [24].

Numerical differentiation can be avoided by using integral isoconversional methods. Integration in Equation (2) leads to
(4)$$g(\alpha )=\frac{A}{\beta }\int_{0}^{T_{\alpha }}\exp\left(-\frac{E}{RT}\right)\;dT$$
where $g(\alpha )=\int_{0}^{\alpha }\frac{1}{f(\alpha )}d\alpha .$ The integral on the right hand side, known as the temperature integral, has no analytical solution and several procedures have been proposed in order to overcome this difficulty. Some approximations have been suggested in order to estimate the value of the temperature integral, but they should be carefully applied because of possible intrinsic inconsistencies, particularly when the activation energy changes with *α* [21,25]. A non-linear method following ICTAC recommendations was proposed by Vyazovkin [26,27]. Numerical integration over small segments of either temperature or time is applied, and the activation energy is obtained by minimization of the function
(5)$$\Phi (E_{\alpha })=\sum_{i=1}^{n}\sum_{j\neq i}^{n}\frac{J_{i}(E_{\alpha })}{J_{j}(E_{\alpha })}$$
where
(6)$$J_{i}(E_{\alpha })=\int_{t_{\alpha_{i}-\Delta \alpha }}^{t_{\alpha_{i}}}\exp\left(-\frac{E}{RT}\right)\;dt$$

In this study, the modified Friedman and the Vyazovkin methods have been applied for computing the activation energy. In order to obtain a model describing the kinetic process, the knowledge of more kinetic parameters is necessary. So the factor $c_{\alpha }=\;Af(\alpha )$ has been computed by using the procedure proposed in [15]. After computing the activation energy, for a fixed value of α, $c_{\alpha }$ can be obtained by using Equation (2)
(7)$$\beta_{i}\frac{d\alpha (T_{\alpha ,i})}{dT}=c_{\alpha }\exp\left(-\frac{E_{\alpha }}{RT_{\alpha ,i}}\right),\;i=1,\ldots ,n$$
and fitting to the experimental data by the least-squares method. After knowing the activation energy *E* and the factor $c_{\alpha },$ for a given heating rate *β*, differential Equation (2) can be solved and so the kinetic parameters can be checked by reconstructing the *α − T* curves for the same heating rates as those used for the runs and comparing these values with those obtained in the laboratory [15,21,28].

## 4. Conclusions

A new hydrothermal route for the synthesis of a diamin iron(III) phosphate is reported. It leads to a crystalline form in the orthorhombic system, space group *Pnma*, at room temperature. The system undergoes a structural phase transition towards monoclinic, space group *P*2_1_/*n*, from 250 to 100 K. The HT-PXRD indicates that the compound is stable up to ca. 440 K. The TG-DTG curves show that three mass losses occur that correspond to the evacuation of one, one third and two thirds of ammonia molecules at 560, 610 and 680 K respectively. The application of the isoconversional methods MFR and Vyaz allowed us to obtain the activation energy in each step as a function of the extent of conversion for both hydrogenated (two samples with different crystal size) and deuterated forms. A similar behaviour of the activation energy was observed for the three compounds. In order to test the kinetic parameters, *α − T* curves were reconstructed and compared with the experimental data, showing a good agreement.

## Figures and Tables

**Figure 1 ijms-21-00781-f001:**
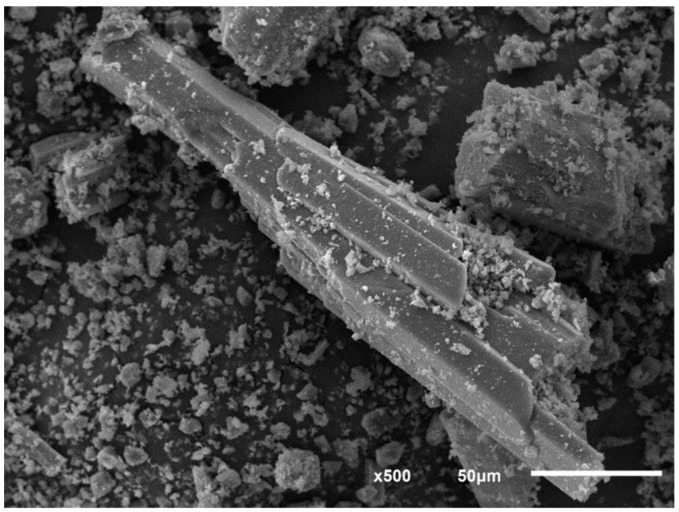
Scanning electron microscopy (SEM) image showing the morphology of Fe(NH_3_)_2_PO_4_ sample.

**Figure 2 ijms-21-00781-f002:**
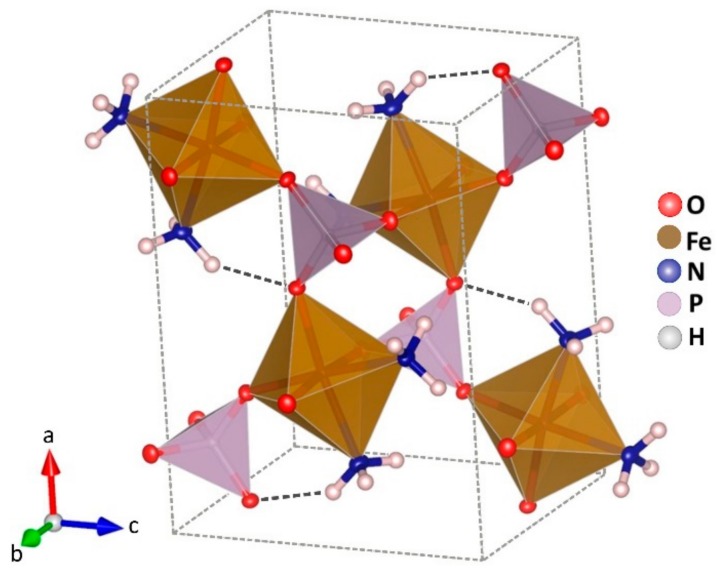
Crystal structure of Fe(NH_3_)_2_PO_4_. Octahedrons FeO_4_N_2_ in brown, and tetrahedrons PO_4_ in violet colour. Atoms of nitrogen in blue and hydrogen in white.

**Figure 3 ijms-21-00781-f003:**
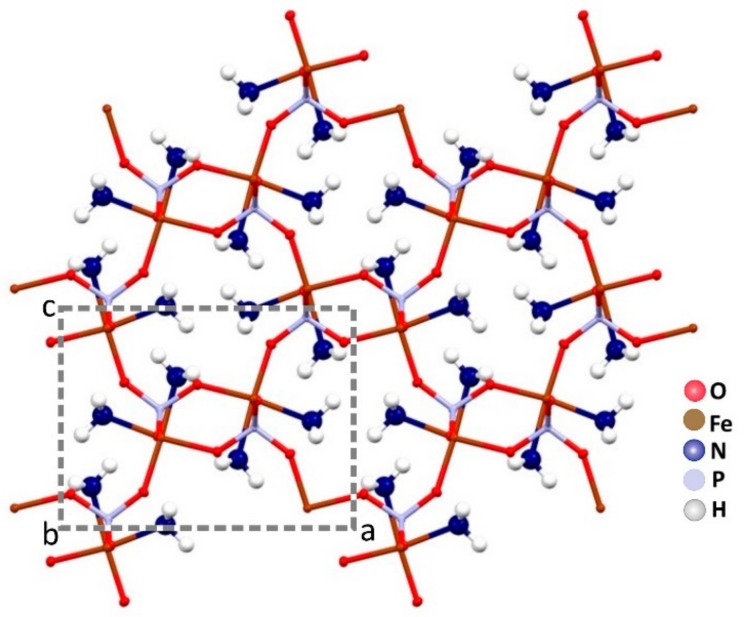
The channels along the *b* axis are fully occupied by the ammonia molecules in the crystal packing of Fe(NH_3_)_2_PO_4_.

**Figure 4 ijms-21-00781-f004:**
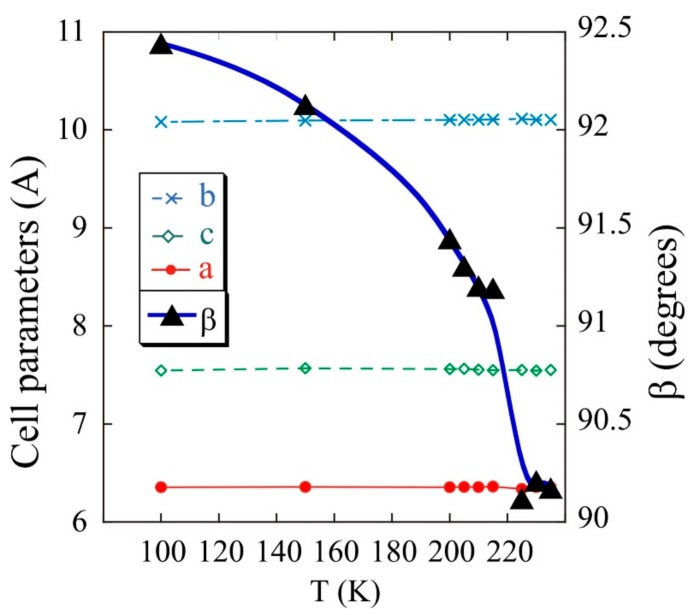
Variation of cell parameters with temperature for Fe(NH_3_)_2_PO_4_.

**Figure 5 ijms-21-00781-f005:**
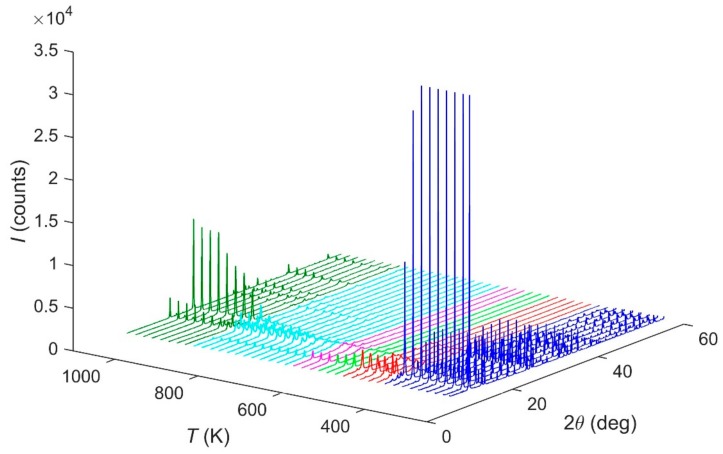
HT-PXRD diffraction patterns evolution of Fe(NH_3_)_2_PO_4_ under thermal treatment.

**Figure 6 ijms-21-00781-f006:**
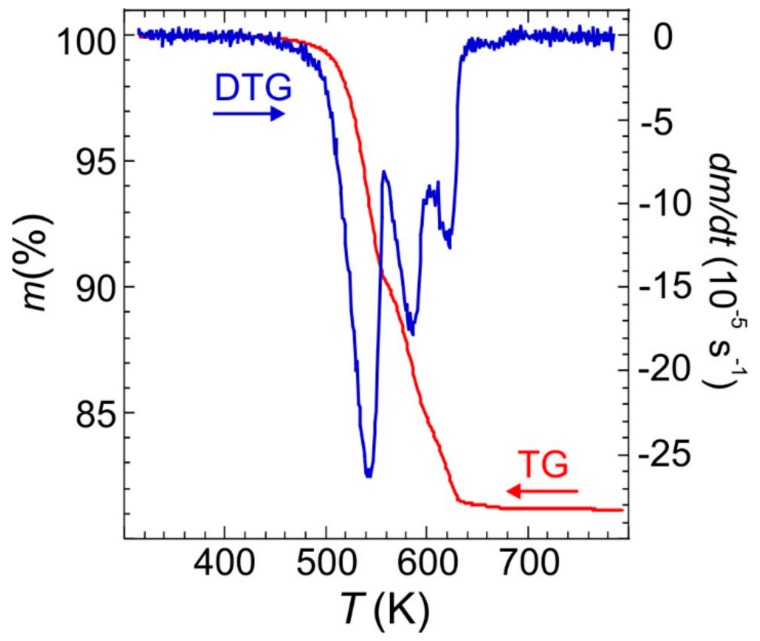
TG and DTG curves of Fe(NH_3_)_2_PO_4_ obtained at 10 K min^−1^ heating rate.

**Figure 7 ijms-21-00781-f007:**
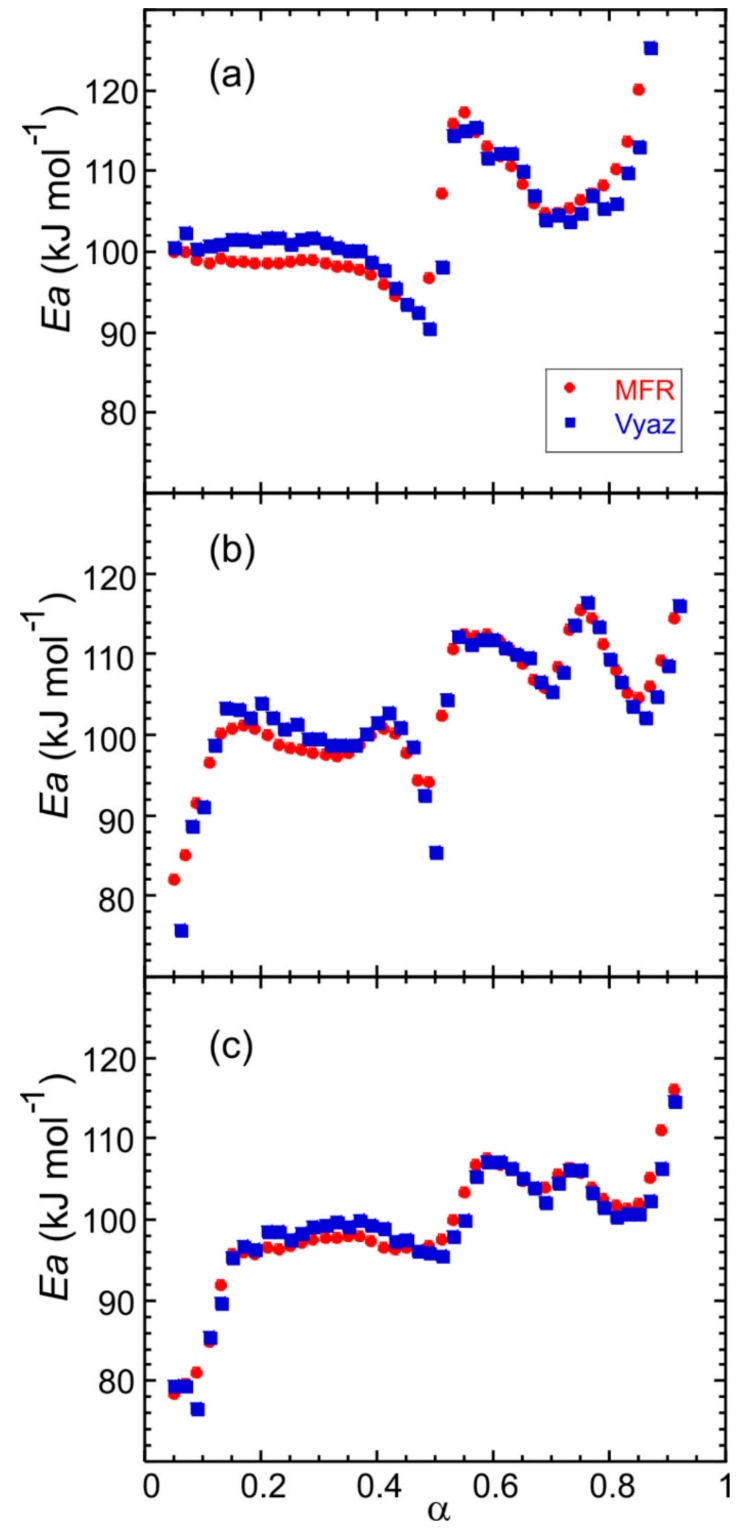
Dependence of the activation energy on the extent of conversion for (a) Fe(NH_3_)_2_PO_4_ (large crystals), and pre-synthesized materials: (**b**) Fe(NH_3_)_2_PO_4_ [11] and (**c**) Fe(ND_3_)_2_PO_4_ [12].

**Figure 8 ijms-21-00781-f008:**
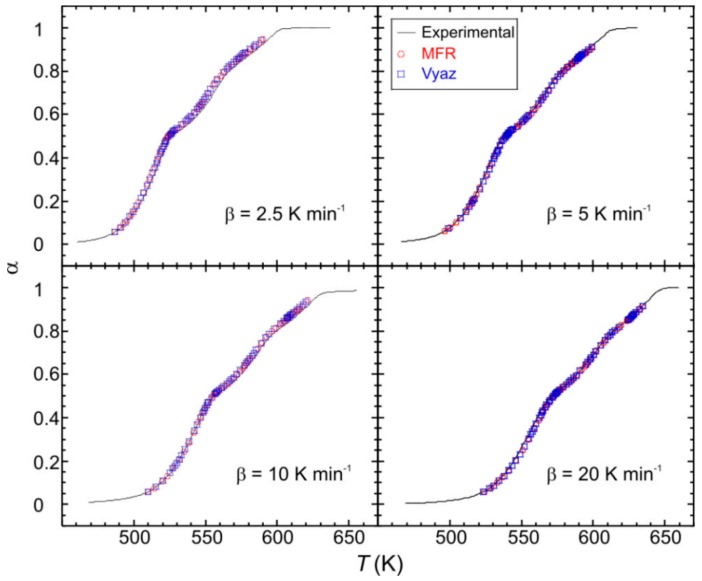
Comparison of the experimental (solid lines) and reconstructed (points) *α − T* curves for the monocrystalline phase.

**Table 1 ijms-21-00781-t001:** Crystal data and structure refinement at room temperature.

Molecular Formula	FeN_2_H_6_PO_4_
Formula mass/g mol^−1^	184.89
Color	Green
Wavelength/Å	1.5418
Crystal system	Orthorhombic
Space group	*Pnma* nº 62
a/Å	10.1116(2)
b/Å	6.3652(1)
c/Å	7.5691(1)
Z	4
Calculated density/g cm^−3^	2.521
Volume/Å^3^	487.165(14)
Crystal size/mm^3^	0.08 × 0.02 × 0.01
θ range/deg	4° to 74°
Absorption coefficient/mm^−1^	27.462
F(000)	372
Index ranges	−12<=h<=12
	−7<=k<=7
	−9<=1<=9
Reflections colleted	4187
Independent reflections	538 [R(int)=0.02]
Completeness θ = 70.00°	100.0%
Refinememt method	Full-matrix least-squares on F^2^
Data/restraints/parameters	538/0/50
Goodness-of-fit on F2	1.143
Final R ind.[I > 2sigma(I)]	R1 = 0.0215
	wR2 = 0.0613
R índices (all data)	R1 = 0.0217
	wR2 = 0.0613
Largest diff. Peak and hole/e Å^−3^	0.289 and −0.816

**Table 2 ijms-21-00781-t002:** Fractional coordinates for the Fe(NH_3_)_2_PO_4_.

Atom	x	y	z	U(eq)
Fe	0.8422(1)	3/4	0.0896(1)	10(1)
P	0.8384(1)	1/4	−0.0412(1)	10(1)
O(1)	0.8348(1)	0.4416(2)	0.0819(1)	16(1)
O(2)	0.7826(1)	3/4	0.3369(2)	16(1)
O(3)	1.0343(1)	3/4	0.1536(2)	16(1)
N(1)	0.6428(2)	3/4	−0.0137(3)	22(1)

**Table 3 ijms-21-00781-t003:** Bond distances [Å] and angles [deg] for the Fe(NH_3_)_2_PO_4_.

Bond lengths (Å)		Bond Angles (deg)	
Fe–O(1)	1.9652(14) × 2)	O(1)–Fe(1)–O(1)	174.45(6)
Fe–O(2)	1.9663(16)	O(1)–Fe(1)–O(2)	90.96(3)
Fe–O(3)	2.0021(15)	O(2)–Fe(1)–O(3)	92.53(3)
		O(2)–Fe(1)–O(3)	93.87(6)
Fe–N(1)	2.1624(19)	O(1)–Fe(1)–N(1)	87.34(3)
Fe–N(2)	2.1661(18)	O(2)–Fe(1)–N(1)	93.34(8)
P–O(1)	1.5354(13) × 2	O(3)–Fe(1)–N(1)	172.79(8)
P–O(2)	1.5318(15) × 2	O(1)–Fe(1)–N(2)	88.90(3)
P–O(3)	1.5427(15) × 2	O(2)–Fe(1)–N(2)	176.68(7)
N(1)–H(11)	0.9612	O(3)–Fe(1)–N(2)	89.45(6)
N(1)–H(12)	0.9913	N(1)–Fe(1)–N(2)	83.34(8)
N(2)–H(21)	0.9534	O(2)–P(1)–O(1)	110.29(5)
N(2)–H(22)	0.9752	O(1)–P(1)–O(1)	105.20(10)
		O(2)–P(1)–O(3)	109.51(10)
		O(1)–P(1)–O(3)	110.74(5)
		P(1)–O(1)–Fe(1)	144.16(8)
		P(1)–O(2)–Fe(1)	144.90(10)
		P(1)–O(3)–Fe(1)	132.56(9)
		Fe(1)N(1)H(11)	109.3
		Fe(1)N(1)H(12)	115.6
		H(11)N(1)H(12)	105.1
		Fe(1)N(2)H(21)	104.8
		Fe(1)N(2)H(22)	124.6
		H(21)N(2)H(22)	106.4
Symmetry transformations used to generate equivalent atoms: #1 x, −y+3/2,z #2 −x+3/2, −y+1,z − 1/2 #3 x, −y + 1/2, z #4 – x + 2, −y + 1, −z #5 −x + 3/2, −y + 1,z + 1/2			

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
