# Peer review of "Kinetic Analysis of the Thermal Decomposition of Iron(III) Phosphates: Fe(NH_3_)_2_PO_4_ and Fe(ND_3_)_2_PO_4"

_ijms, 2020, doi:10.3390/ijms21030781_

Round 1

Reviewer 1 Report

In the manuscript entitled “Kinetic analysis of the thermal decomposition of iron(III) phosphates: Fe(NH3)2PO4 and Fe(ND3)2PO4” the authors describes an optimization of their previously reported hydrothermal synthesis of the diamin iron phosphate Fe(NH3)2PO4. Applying the new synthetic conditions the desired material can be produced as large single crystals, instead of the previously obtained polycrystalline phase. The X-ray structure of a single crystal has been obtained and the activation energies of the thermal decomposition of the new material, as well as those of Fe(NH3)2PO4 and Fe(ND3)2PO4 previously obtained, have been evaluated through thermogravimetric experiments.

In the opinion of this referee the manuscript is suitable to be published on IJMS, after the revisions below reported.

In view of the fact that the authors report an optimization of an old hydrothermal synthesis for Fe(NH3)2PO4 (Hydrothermal synthesis of iron(III) phosphates in the presence of urea. J Chem Soc Dalton Trans. 2000, 5, 787-90, reference 10 in the manuscript), in the “introduction” or also in the “conclusion” the developing potentiality of this specific material should be described in more detail. Some data concerning the main differences between the new adopted and the previously reported hydrothermal routes for the synthesis of the diamin iron phosphate should be described and discussed in the section “results”. For the sake of clarity the method to convert the thermogravimetric data to those reported in figure 8 (the lines shown as experimental) should be briefly described. The manuscript should be accurately revised. E.g.: row 18 “bellow” should be below; rows 36-40 the sentence is too long and should be rephrased; row 41 “molar ratio” does it mean molar ratio between reactants? rows 122 the sentences “…the evacuation of one ammonium molecule.” is not clear. Does it mean one ammonium molecule per formulae? Row 132 in the sentence “..as a function of the extent of conversion with an increment Δα = 0.01” α should be defined; row 220 “ecuation” should be equation; etc…

Author Response

Response to Reviewer 1 Comments

In the manuscript entitled “Kinetic analysis of the thermal decomposition of iron(III) phosphates: Fe(NH3)2PO4 and Fe(ND3)2PO4” the authors describes an optimization of their previously reported hydrothermal synthesis of the diamin iron phosphate Fe(NH3)2PO4. Applying the new synthetic conditions the desired material can be produced as large single crystals, instead of the previously obtained polycrystalline phase. The X-ray structure of a single crystal has been obtained and the activation energies of the thermal decomposition of the new material, as well as those of Fe(NH3)2PO4 and Fe(ND3)2PO4 previously obtained, have been evaluated through thermogravimetric experiments.

Point 1: In view of the fact that the authors report an optimization of an old hydrothermal synthesis for Fe(NH3)2PO4 (Hydrothermal synthesis of iron(III) phosphates in the presence of urea. J Chem Soc Dalton Trans. 2000, 5, 787-90, reference 10 in the manuscript), in the “introduction” or also in the “conclusion” the developing potentiality of this specific material should be described in more detail.

Response 1: Potential applications of this specific material have been included in the introduction (for instance, see Lin, K.; Wu, Ch.; Chang, J. Advances in synthesis of calcium phosphate crystals with controlled size and shape. Acta Biomaterialia 10 (2014) 4071–4102).

Point 2: Some data concerning the main differences between the new adopted and the previously reported hydrothermal routes for the synthesis of the diamine iron phosphate should be described and discussed in the section “results”.

Response 2: We have included some explanations of the new hydrothermal route at the beginning of the Morphological and Structural Results Morphological subsection. Data concerning the main differences between both diamine iron phosphates (crystal size, morphology, and activation energy) can be observed comparing the results of this paper with reference Hydrothermal synthesis of iron(III) phosphates in the presence of urea. J Chem Soc Dalton Trans. 2000, 5, 787-90, [10] in the manuscript.

Point 3: For the sake of clarity the method to convert the thermosgravimetric data to those reported in figure 8 (the lines shown as experimental) should be briefly described.

Response 3: In the revised version of the manuscript we have included some sentences on lines 208-211, in order to clarify how thermogravimetric data are converted to those reported in Figure 8.

Point 4: rows 36-40 the sentence is too long and should be rephrased;

Response 4: It was rephrased.

Point 5: row 41 “molar ratio” does it mean molar ratio between reactants?

Response 5: Yes, it does. The text has been modified.

Point 6: rows 122 the sentences “…the evacuation of one ammonium molecule.” is not clear. Does it mean one ammonium molecule per formulae?

Response 6: Yes, it does. The text has been modified.

Point 7: Row 132 in the sentence “..as a function of the extent of conversion with an increment Δα = 0.01” α should be defined;

Response 7: The definition of α has been included at the beginning of section 3.3.

Point 8: The manuscript should be accurately revised. E.g.: row 18 “bellow” should be below;

row 220 “ecuation” should be equation; etc…

Response 8: The manuscript was revised and corrected.

Reviewer 2 Report

The paper- Kinetic analysis of the thermal decomposition of 3 iron(III) phosphates: Fe(NH3)2PO4 and Fe(ND3)2PO4, presents the obtaining and characterization of Fe (NH3) 2PO4 as compared to the deuterated compound.

The paper is well presented with clear explanations. The data obtained are supplemented with XRD studies and thermoanalytical data. Thermal analysis studies are supplemented with XRD data obtained at different temperatures. The kinetic study is performed using the Friedman Modified and Vyazovkin method. The data are clearly presented. The only recommendation would be that for a better understanding, the chapter of materials and methods should be placed before the results and discussions chapter.

Author Response

The paper is well presented with clear explanations. The data obtained are supplemented with XRD studies and thermoanalytical data. Thermal analysis studies are supplemented with XRD data obtained at different temperatures. The kinetic study is performed using the Friedman Modified and Vyazovkin method. The data are clearly presented.

Point 1: The only recommendation would be that for a better understanding, the chapter of materials and methods should be placed before the results and discussions chapter.

Response 1: This recommendation is possible, but we have tried to follow the usual format of this journal.
